# Effects of Incorporating Caramel, Carrot, and Tomato Powder on the Quality Characteristics of Soy Protein-Based Meat Patties

**DOI:** 10.3390/foods13142224

**Published:** 2024-07-15

**Authors:** Xinyu Shi, Zun Wang, Zhongxiang Fang

**Affiliations:** School of Agriculture, Food and Ecosystem Sciences, Faculty of Science, The University of Melbourne, Parkville, VIC 3010, Australia; xishi2@student.unimelb.edu.au (X.S.); zunw@student.unimelb.edu.au (Z.W.)

**Keywords:** soy protein, meat analogs, plant-based meat patty, natural pigment, antioxidant activity

## Abstract

Plant protein-based foods have become dietary preferences worldwide, and the quality of this food group is highly associated with the ingredients used. This study investigated the effects of incorporating caramel, tomato powder, and carrot powder on the product quality of soy protein-based meat patties (SPMPs). The color, total phenolic content (TPC), antioxidant activity, lipid oxidation, and texture profile of uncooked and cooked soy protein meat patties were analyzed. Among the cooked SPMPs, caramel SPMPs exhibited the lowest color difference (ΔE) values, and the ΔE value of tomato SPMPs was lower than that of carrot SPMPs, indicating that caramel has the best color stability, and the tomato experienced less color change than the carrot during cooking. Notably, carrot SPMPs exhibited lower color stability during refrigeration storage than the others. Both carrot and tomato powders increased the total phenolic content (TPC) and antioxidant stability and inhibited lipid oxidation in SPMPs during cooking. However, tomato SPMPs exhibited higher TPC values and greater antioxidant stability compared to carrot SPMPs. The addition of caramel and carrot powders decreased the hardness of raw SPMPs, but tomato powder increased the hardness. The texture profile of tomato SPMPs was more affected by the cooking process compared to caramel and carrot SPMPs. This study showed that incorporating both carrot and tomato powders positively influenced the quality characteristics of SPMPs compared to caramel powder, however, tomato powder exhibited superior efficacy.

## 1. Introduction

Animal food products account for a high proportion of the Western diet, and high meat consumption has been widely debated recently due to its adverse impacts on the environment, animal welfare, and human health [1]. Meat production is associated with several concerns, including excessive use of water resources and land, greenhouse gas emissions, as well as detrimental effects on aquatic and terrestrial biodiversity and animal welfare [2]. Livestock farming significantly impacts the environment, contributing approximately 14.5% of human-induced greenhouse gas emissions (GHGs), it also consumes nearly 40% of arable land and 56% of agricultural freshwater resources [3]. Eshel et al. suggested that replacing meat with plant-based alternatives could potentially reduce national dietary land use by 34%, nitrogen fertilizer by 47%, and GHG emissions by 38% in the USA [4]. Moreover, prolonged consumption of red or processed meat has been associated with the development of cardiovascular diseases and type 2 diabetes [2]. Concerns also extend to additional health risks such as colon cancer and breast cancer in women, which may be exacerbated by the use of hormones in meat production to enhance growth rates and yields [5]. For these reasons, there has been a shift toward a more plant-based diet for increased sustainability, leading both the food industry and researchers to actively seek plant proteins as alternatives for animal-sourced proteins. The market for plant protein-based foods is expanding rapidly, driven by an increase in consumers following vegan, vegetarian, or flexitarian lifestyles [6]. As part of this trend, flexitarian consumers expect that consuming plant-based products will help them lower their intake of cholesterol and saturated fatty acids associated with meat consumption [7]. Texturized vegetable protein is widely recognized for its significant role in addressing two prominent food trends: the growing demand for high-quality, low-fat foods, and the expanding market of functional and nutraceutical foods [7]. Soy-based texturized vegetable protein, as a meat substitute, offers various economic and functional advantages. It is a plant-based protein product characterized by low saturated fat and calories, a rich concentration of essential amino acids, and is cholesterol-free [8], with meat analogs having 96% less saturated fatty acids, 58% less energy, 33% less sodium, no cholesterol, and double the amount of fiber than a meat burger [9].

The term “meat analogs” refers to a group of plant protein-based products closely resembling animal whole-muscle meat in terms of appearance, texture, and flavor, as well as some restructured products that resemble processed meats, such as sausage and patties [10]. It is also known as meat substitutes, meat alternatives, mock meat, or imitation meat [11]. The major plant protein ingredients in the production of plant-based meat analogs are soy protein, pea protein, and wheat gluten [11]. Although the processing technology of plant-based meat analogs has been well developed, differences still exist between meat analogs and traditional meat products in terms of appearance, taste, flavor, and texture [12]. Among these challenges, the appearance of plant-based meat analogs, particularly the color, is a significant issue for researchers. Color plays a vital role in the sensory experience for meat analogs because the initial sensory experience of food starts with the sense of sight, which is crucial in shaping consumer expectations and influencing acceptance [13].

Caramel color is one of the world’s most widely used food pigments, with strong coloring capacity and excellent water solubility [14]. Typically, caramel color with heat stability can provide a brown color appearance to the final meat analog [15]. It has been used in meat analog products variously, such as vegetarian meat substitutes, e.g., Bacon Bits from Augason Farms, Frieda’s plant-based Soyrizo™, and BOCA’s All American Veggie Burgers. Tomato powders are highly nutritious, providing essential nutrients to humans such as folate, vitamin C, and potassium. They are also rich in carotenoids, with lycopene being the most abundant; carotenoids contribute to the antioxidant activity of tomatoes, making them valuable sources of antioxidants [16]. Additionally, tomatoes contain vitamin E, trace elements, flavonoids, phytosterols, and various water-soluble vitamins, further enhancing their health benefits [16]. Carrot powder rich in carotenoids, flavonoids, polyacetylenes, minerals, and vitamins, offers a wide array of health and nutritional benefits. The carotenoids, polyphenols, and vitamins found in carrots serve as antioxidants, anticarcinogens, and immune enhancers, contributing to their significant health-promoting properties [17]. Recently, natural pigments were experimented with in some meat analogs to mimic the color of real meat. For example, Sakai et al. [18] developed a browning system for plant-based meat analog patties using sugar beet pectin and laccase to enhance the product appearance, making it more similar to real meat patties. Milani and Conti [19] used caramel color in textured soy protein to improve the sensory quality of meat analog and soy burgers. Carrot juice extract was used in the veggie burgers from “Morning Star Farms™”, and Aksu and Turan [20] also used carrot extract to improve the storage quality of vacuum-packaged fresh meat products. Lyu et al. [21] investigated the effect of tomato powder in soy protein-based high-moisture meat analogs, and Savadkoohi et al. [22] incorporated tomato pomace with soy protein isolate to produce a meat-free sausage. However, research on the effect of natural pigments on both the color and nutritional quality of meat analogs is still limited. Therefore, this research aimed to investigate the impacts of three colorants (caramel, carrot, and tomato powder) on the color and pH of soy protein meat analogs during refrigerated storage. The color, total phenolic content (TPC), antioxidant activity, lipid oxidation, and texture profile of the raw and cooked soy protein meat analogs were also compared. This research may provide a reference for the processing of plant protein-based meat products using natural colorants to mimic the real meat color, with potential health benefits.

## 2. Materials and Methods

### 2.1. Materials

Uncolored texturized soy protein (TSP) was obtained from Bob’s Red Mill Co. Ltd. (Milwaukie, OR, USA). Wheat gluten flour (Vegan Grocery Store Pty. Ltd., Melbourne, Australia), food-grade methylcellulose (Delices d‘Oz Pty. Ltd., Port Augusta, SA, Australia), coconut oil (Woolworths Group Ltd., Mel, VIC, Australia), and table salt (Woolworths Group Ltd., Mel, VIC, Australia) were purchased. Caramel powder, whole tomato powder, and carrot powder were obtained from Maple Lifesciences Co. Ltd. (Haryana, India), Spice Masters Australia (Kingsgrove, NSW, Australia), and Opera Foods Pty. Ltd. (Warners Bay, NSW, Australia), respectively. All chemicals were purchased from Sigma-Aldrich (Castle Hill, NSW, Australia).

### 2.2. Sample Preparation

#### 2.2.1. Preparation of Soy Protein Meat Patties (SPMPs)

The meat patties were made from TSP (25.8%), boiled water (54.2%), wheat gluten flour (12%), coconut oil (6%), methylcellulose (1%), and salt (1%) [23]. Briefly, the uncolored dry TSP was immersed in boiling water (1:6, *w*/*v*) for 15 min for rehydration. The excess water was squeezed out from the swollen TSP to achieve a TSP-to-water ratio of 1:2.1. Then the rehydrated TSP was mixed thoroughly with the coconut oil first and other ingredients. The uncolored SPMPs were used as the control group, and another nine colored SPMPs were prepared with the addition of caramel, carrot, and tomato powder by different weight percentages (Table 1), based on our preliminary experiments on the color similarity of SPMPs to real meat. Finally, the formulated mixtures were molded to a diameter of 6.5 cm, a thickness of 1.2 cm, and a 50 g patty. Three independent batches were prepared.

The SPMPs were wrapped with plastic wrap, packaged in sterile airtight bisphenol A-free bags, and stored at 4 °C for 0, 1, 4, 7, and 10 days for cooking and analysis.

#### 2.2.2. Cooking Condition

The SPMPs were cooked individually in a frypan at approximately 170 °C and turned over every 1 min until the internal temperature reached 75 °C by inserting a digital thermometer (NORONIX 600 N 50 °C to + 200 °C, Mitchell Park, Australia) into the geometric center of the patties. Then, the patties were taken out of the frypan, cooled to room temperature, and covered with plastic wrap for further analysis.

### 2.3. Color Measurement

The color of the SPMPs within 10 days of storage was measured before and after cooking by using a Minolta Colorimeter with a C illuminant and 8 mm aperture size (Model CR-400, Minolta Co., Ltd., Osaka, Japan) [24]. The instrument was calibrated with a standard white tile (L* = 93.80, a* = 4.84, b* = −0.80) before use and the CIELAB color system for L* (lightness), a* (redness), and b* (yellowness) values were recorded. Each patty was measured at six different locations. The color difference (ΔE) was calculated to describe the changes in the color of the raw SPMPs and cooked SPMPS, and the color change during the 1, 4, 7, and 10 days of storage (4 °C) compared to day 0, which was calculated using the following formula [22]:(1)$$\Delta E=\sqrt{{{\Delta L}^{*}}^{2}+{{\Delta a}^{*}}^{2}{{+\Delta b}^{*}}^{2}}$$

### 2.4. pH Measurement

Three grams of SPMP sample were homogenized with 27 mL of milli-Q water using an IKA-T25 Ultra-Turrax homogenizer (IKA-Werke GmbH & Co, Staufen, Germany) at 8400 rpm for 2 min in an ice water bath [25]. A pH meter (HI1230, Hanna Instruments, Smithfield, RI, USA) was used to measure the pH of the resulting homogenate at room temperature. All tests were performed in triplicate.

### 2.5. Total Phenolic Content (TPC)

The TPC of raw and cooked SPMPs was analyzed using a modified Folin–Ciocalteu assay [26]. Briefly, 25 mL of methanol and a 5 g SPMP sample were mixed and shaken for one hour on a platform mixer (Ratek Instruments Pty Ltd., Boronia, Australia). After centrifugation of the mixture at 4000× *g* for 10 min at 4 °C, the supernatant was collected and filtered with a Whatman No. 1 filter paper; 0.1 mL of filtrate and 0.5 mL of Folin–Ciocalteu solution were mixed for 10 min. The new mixture was added with 0.4 mL of 7.5% sodium carbonate solution, followed by a vortex and incubation for 30 min at room temperature in the dark. The absorbance of the resulting solution was measured at 760 nm with a UV-Vis spectrophotometer (Multiskan GO, Thermo Scientific, Vantaa, Finland). Gallic acid (0.05–0.25 mg/mL) was used as a standard. The TPC was expressed as mg gallic acid equivalent (GAE)/100 g sample. All tests were performed in triplicate.

### 2.6. DPPH Radical Scavenging Activity

The DPPH (2,2-diphenyl-1-picrylhydrazyl) radical scavenging assay was used to estimate the antioxidant activity of the SPMPs [27]. To prepare the DPPH stock solution, 24 mg of DPPH was dissolved in 100 mL methanol and stored in the dark at −20 °C. The DPPH working solution (absorbance of 1.1 ± 0.02 units at 517 nm) was prepared freshly by diluting about 8 mL of stock solution in 40 mL of methanol [28]. An aliquot of 1 mL of the above-filtrated sample and 4 mL of fresh DPPH working solution were mixed and kept in the dark for 30 min. The absorbance of the resulting solution was measured at 517 nm with the Multiskan GO UV-Vis spectrophotometer. Trolox (12.5–100 mg/L), methanol, and the DPPH working solution were used as the standard, blank, and control, respectively. All tests were performed in triplicate. Radical scavenging activity (RSA (%)) was calculated using the following formula:(2)$$RSA\;\left(\%\right)=\frac{Absorbance\;of\;control-Absorbance\;of\;sample}{Absorbance\;of\;control}\times 100$$

### 2.7. Lipid Oxidation (TBARS Assay)

Lipid oxidation of the raw and cooked SPMP samples was analyzed based on the 2-thiobarbituric acid reactive substances (TBARS) assay, according to the method of Xiong et al. (2022), with some modifications. About 5 g of SPMP sample in 15 mL of 10% trichloroacetic acid (TCA) solution was homogenized using a homogenizer (Polytron PT 10–35 GT, Kinematica AG, Luzern, Switzerland) at 8400 rpm for 60 s. The mixture was centrifuged at 3500× *g* and 4 °C for 8 min. Two milliliters of supernatant were collected and mixed with 2 mL of 20 mM 2-thiobarbituric acid (TBA) solution and incubated at 95 °C for 30 min in a water bath. Then the sample was cooled to room temperature in cold water and centrifuged at 3000× *g* 4 °C for 10 min [29]. The absorbance of the supernatant was measured at 532 nm with the Multiskan GO UV-Vis spectrophotometer. The chemical 1,1,3,3-tetraethoxypropane (0–20 μM) was used as the standard. The TBARS value was expressed as a milligram malondialdehyde equivalent per kilogram (mg MDA/kg) sample. All tests were performed in triplicate.

### 2.8. Cooking Loss

After cooking the samples as described in Section 2.2.2, the proportion of fluid lost, which includes water, lipids, proteins, and minerals, was defined as the cooking loss and calculated using the formula below [30]:(3)$$Cooking\;loss\left(\%\right)=\frac{raw\;patty\;weight-cooked\;patty\;weight}{raw\;patty\;weight}\times 100$$

### 2.9. Texture Profile Analysis

The texture profile analysis (TPA) of the SPMPs was conducted by a double-bite compression test using a single-column universal testing machine (Lloyd LS5 universal testing machine, Ametek Inc., Berwyn, PA, USA), fitted with a cylindrical probe (20 mm diameter), as described by Xiong, Zhang, Warner, Hossain, Leonard, and Fang [25], with modifications. The raw and cooked samples were cut into square shapes in dimensions of 15 × 15 × 10 mm. The raw and cooked sample was double-compressed to 50% of the original height at a constant speed of 1 mm/s and 1 s time interval between the two cycles. Nine replicates (*n* = 9) were performed for each sample. The TPA parameters of hardness, cohesiveness, gumminess, springiness, and chewiness were calculated using the machine software.

### 2.10. Statistical Analysis

All data were expressed as mean ± standard deviation (SD), and one-way analysis of variance (ANOVA) was conducted using Minitab software (Minitab Windows, version 17, Sydney, Australia). Tukey’s post hoc test at a 95% confidence level was used to compare the differences between different SPMP formulations, and the T-test (*p* < 0.05) was used to compare raw and cooked samples in the same formulation.

## 3. Results and Discussion

### 3.1. Color Measurement

The surface color of the meat analog is an important visual indicator of its quality. The visual observation and color values (L*, a*, b*, and ΔE) of the raw and cooked SPMPs are shown in Figure 1 and Table 2, respectively. The L* value (lightness) of all SPMPs was significantly decreased after cooking (*p* < 0.05). The reduced lightness could be explained by Maillard reactions during the cooking process [31], as shown in Figure 1. The a* (redness) and b* (yellowness) values of TSP (control group) increased dramatically after the cooking process (*p* < 0.05). Compared to other colorants, the increased addition of caramel powder resulted in a distinct decrease in lightness and yellowness (*p* < 0.05) in both raw and cooked SPMPs. Caramel powder is usually obtained by caramelization of sugar, which produces dark-brown color hues [32]. As shown in Figure 1, with the amount of caramel powder increasing, the raw and cooked SPMPs turned to a “burnt” appearance, similar to cooked real meat products. Both raw carrot and tomato SPMPs resulted in significant increases in a* and b* values compared to the TSP. Compared with tomato powder SPMPs, carrot powder ones presented higher yellowness (*p* < 0.05), which could be explained by the presence of β-carotene in the carrot powder. However, the yellowness of all carrot SPMPs decreased significantly after cooking. The value of the total color difference (ΔE) indicates the color stability of samples after the cooking, where a lower ΔE represents a higher color stability. Tomato SPMPs have relatively lower ΔE values compared to carrot SPMPs, suggesting that during the cooking process, tomato SPMPs experienced less color change and had higher color retention than the carrot SPMPs. Caramel SPMPs exhibited the lowest ΔE values among all samples, indicating the highest degree of color stability. This finding is in agreement with Kaur and Sogi [33], who reported a decreased pigment content of carrots after thermal treatments due to the degradation of carotenoids. The color changes of tomato SPMPs (increased a* and b* values but decreased L* value) were consistent with the previous research of Kapituła [34] using tomato derivatives in meat products, which was attributed to the high content of lycopene in tomato derivatives.

Table 3 shows the changes in L*, a*, b*, and ΔE values during the 10-day refrigerated storage (4 °C). All SPMPs had no obvious change in lightness during the period. In terms of redness, only the tomato group showed a declining trend, possibly due to the poor stability of lycopene in tomato powder, which is sensitive to oxygen, light, high temperature, catalyst, acids, and metal ions [35]. The yellowness of carrot SPMPs also decreased with the extension of the storage, which was in agreement with the results of Yadav et al. [36] and explained by the degradation of β-carotene. After 4 days of refrigeration storage, carrot SPMPs demonstrated relatively higher ΔE values compared to the other SPMPs, signifying poorer color stability of carrot SPMPs during refrigeration storage. This observation suggests that carrot SPMPs underwent more pronounced color changes during refrigerated storage than the other SPMPs.

### 3.2. pH Measurement

Table 4 shows the pH changes of raw SPMPs during the 10 days of refrigeration storage (4 °C), where the mean values of pH ranged from 4.95 to 6.72. The pH values of TSP showed a continuous decline from 6.72 to 6.21. Liu et al. [37] suggested that the pH of the soy-based meat analog decreased during storage as it was susceptible to microbial growth due to its high moisture and protein content. Different concentrations of caramel powder had very little effect on the pH values. In the carrot group, the increasing addition of carrot powder resulted in a decline in the pH of SPMPs. Aksu and Turan (2022) also reported the pH-reducing effect in carrots due to trace amounts of succinic acid and lactic acid and suggested that the low pH environment could suppress the bacteria growth and oxidation of protein and lipids, resulting in a longer shelf life. The addition of tomato powder showed a significant reduction in the pH value of SPMPs, where 15% tomato patty resulted in the lowest pH values (*p* < 0.05) among all groups during refrigeration storage. This could be caused by the natural presence of citric and malic acids in the tomato powder. Østerlie and Lerfall (2005) also observed that the acidic characteristics of tomato products reduced the pH values of meat products, which resulted in lower microbial growth, enhanced storage stability, and provided a better color and potential health advantages.

### 3.3. Total Phenolic Content (TPC)

The TPC of raw ingredients was tested. As the base material, the TSP and wheat gluten flour contained 27.35 and 40.19 GAE/100 g of TPC, respectively. The TPC values of caramel, carrot, and tomato powder were 24.75, 121.74, and 298.77 mg GAE/100 g, respectively. The TPC of tomatoes was over double that of carrots, which is consistent with the findings of Bozalan and Karadeniz [38] and Katırcı et al. [39] that tomatoes have a much higher TPC compared to carrots.

The TPC values of raw and cooked SPMPs are shown in Table 5. There was no significant difference between the raw and cooked TSP compared to those of caramel SPMPs in TPC values, which could be explained by the low TPC of caramel powder. However, incorporating 10% and 15% carrot powder led to a significant rise in the TPC values of both raw and cooked carrot SPMPs (*p* < 0.05). The difference in TPC between raw and cooked carrot SPMPs was not significant, indicating that the phenolic compounds in carrot SPMP patties were relatively thermally stable. Chlorogenic acid was the predominant phenolic compound in most carrot varieties [40], of which, the degradation temperature was up to 200 °C [41]. Similar to carrot powder, an increase in the concentration of tomato powder led to higher TPC in the tomato meat patties (*p* < 0.05). This finding aligns with the studies of Wójtowicz et al. [42] and Yagci et al. [43], which reported that the incorporation of tomato pomace increased the TPC of snack foods. At equivalent addition levels, the TPC of tomato SPMP surpassed that of carrot SPMP because of the higher TPC (298.77 mg GAE/100 g) in tomato powder as discussed above. This accounted for the highest TPC observed (85.86 mg GAE/100 g) in the SPMPs with 15% tomato powder added, compared to all other patties. There was no significant difference in TPC between raw and cooked tomato SPMPs, which is consistent with the study by Dewanto et al. [44] regarding heat-processed tomatoes. The predominant phenolic compound in the tomato is caffeic acid [45], which is subjected to decomposition at about 156 °C [46]. Given the cooking temperature (170 °C) utilized in this study, the unobvious change in TPC values may be attributed to the limited distribution of caffeic acid on the meat patty surface; the center temperature of the cooked patties was only about 75 °C.

### 3.4. DPPH Radical Scavenging Activity

The DPPH assay is a rapid, widely used, simple, and cost-effective method to estimate the antioxidant activity of foods [47]. The DPPH radical scavenging activity (RSA%) of raw ingredients of the SPMPs was measured and ranked from low to high as follows: wheat gluten (31.23%), TSP (34.35%), caramel powder (47.26%), carrot powder (88.21%), and tomato powder (298.77%).

Table 5 shows the RSA% of raw and cooked SPMPs, where the RSA% of TSP and caramel-added SPMPs increased after cooking. Song et al. [48] reported similar results (the DPPH RSA% of meat analog increased with the heating time at 95 °C), suggesting that the promoted antioxidant activity could come from the Maillard reaction products. Compared to TSP, the addition of carrot and tomato powders resulted in a concentration-related increase in the RSA% in the meat patties, whether in the raw or cooked products, while there was no change with the addition of caramel (Table 5). Phenolic compounds are terminators of free radicals, and high TPC is highly related to the ability to scavenge DPPH radicals [49], so the results were as expected, i.e., the increased concentration of tomato and carrot powders in the SPMPs resulted in a higher TPC and led to a higher radical scavenging activity. Wójtowicz et al. (2018) also observed increased antioxidant activity in functional snacks with the addition of freeze-dried tomatoes after cooking, and the rich content of lycopene and phenolic compounds in tomatoes might have contributed to this phenomenon. In addition, the study by Sam et al. [50] showed that incorporating carrot pastes in frankfurters increased the TPC and antioxidant activities due to the presence of carotenoids, polyphenols, ascorbic acid, and polyacetylenes.

### 3.5. Lipid Oxidation (TBARS Assay)

TBARS assay is a method used to detect lipid oxidation by measuring malondialdehyde [29]. The TBARS values of the raw and cooked SPMPs with different formulations are presented in Table 5, where the values for all SPMPs significantly increased after cooking. There is no noticeable difference between the TSP (control group) and the caramel group in both raw and cooked SPMPs (*p* > 0.05). However, the TBARS values in the raw SPMPs incorporating carrot and tomato significantly increased, aligning with the increased amount, which was much higher than those of the TSP and caramel groups. Similarly, Kapituła [34] reported a positive relationship between TBARS value and adding tomato powder in meatloaves, suggesting that the carotenoids in the tomato powder could perform as either an antioxidant or a pro-oxidant, depending on factors such as storage conditions, concentration, the type of carotenoid, and constitution of the system. The increased lipid oxidation by the addition of tomato-derived ingredients was also reported by Deda et al. [51] in frankfurter sausage, but the mechanisms need further investigation.

Although adding tomato and carrot powders increased the TBARS values in the raw meat patties, it resulted in less of an increase (<1.5 mg MDA/kg) in TBARS values after cooking compared with the TSP and caramel groups (about 2 mg MDA/kg). The results indicated that tomato and carrot powders could inhibit lipid oxidation during cooking. Domínguez et al. [52] reported a significant decrease in the lipid oxidation of cooked sausages with the addition of tomato paste and tomato powder. Kim et al. [53] suggested that the antioxidant activity of lycopene in tomato powder could influence the lipid oxidation of low-fat pork sausages. Similarly, carrots decreased the lipid oxidation of SPMP during the cooking process, because of the presence of various compounds with antioxidant potential, such as high concentrations of carotenoids and phenolic compounds [20,54].

### 3.6. Cooking Loss

Cooking loss reflects the extent of meat shrinkage during cooking, which is an indicator of cooking yield and the juiciness of the final meat product [18]. Table 6 shows the cooking loss (%) of the SPMPs in different formulations. A decreasing trend in the cooking loss was observed in tomato SPMAs with the increased amount of tomato powder, which could be due to the high amount of dietary fiber (9% of whole tomato powder; data are from the product labeling) in the tomato powder. Savadkoohi, Hoogenkamp, Shamsi, and Farahnaky [22] reported that tomato pomace has a high water absorption ability, due to the cellulose-based polysaccharides and pectin that absorb and entrap water. Sakai, Sato, Okada, and Yamaguchi [18] also reported that incorporating dietary fiber in meat products enhanced the water- and oil-holding capacity and decreased the cooking loss of the cooked meat products. However, the addition of caramel and carrot powders did not show any differences in the cooking loss of the SPMAs (Table 6).

### 3.7. Texture Profile Analysis

Table 6 shows the texture parameters of hardness, cohesiveness, springiness, and chewiness of the raw and cooked SPMPs. In texture analysis, hardness is the maximum force during the initial compression, cohesiveness is determined by dividing the area of work during the second compression by the area of work during the initial compression, springiness is determined by dividing the distance of the second compression by the distance of the initial compression, and the chewiness is obtained by multiplying hardness, cohesiveness, and springiness [18]. In the raw SPMPs, the addition of caramel and carrot pigments resulted in a decreased hardness (*p* < 0.05). On the contrary, the addition of tomato powder increased (*p* < 0.05) hardness, with no discernible impact from the concentration. TSP as the control group has the highest cohesiveness (*p* < 0.05) in both raw and cooked SPMPs. For the springiness of the raw SPMPs, there were no significant differences between all formulations. The incorporation of colorants generally decreased the cohesiveness and chewiness of the raw SPMPs compared with the TSP, but not the springiness (*p* > 0.05). After cooking, the tomato SPMPs still had the highest hardness (*p* < 0.05). The cohesiveness and chewiness of all SPMPs were enhanced after cooking (*p* < 0.05). Moreover, an increasing trend in springiness, along with a decline in cohesiveness and chewiness, was observed in the cooked tomato SPMPs, with increased addition of tomato powder. Compared with tomato powder, the effect of the concentration of caramel and carrot powders on the texture characteristics was less pronounced in the cooked SPMPs. The results implied that as the tomato powder content increased, the SPMPs tended to be more elastic and easier to chew.

The influence of tomato powder or pomace on the texture profile of plant-/animal-based meat products was widely reported. Lyu, Ying, Zhang, and Fang [21] found that the increased amount of whole tomato powder resulted in increased hardness, decreased cohesiveness and chewiness, and unchanged springiness in the raw extruded meat analogs. They suggested that the insoluble fiber of tomato powder not only contributed to the hardness but also disrupted the protein cross-linking, resulting in the demoted cohesiveness and chewiness. Similarly, Savadkoohi, Hoogenkamp, Shamsi, and Farahnaky [22] reported that the addition of tomato pomace increased hardness but decreased cohesiveness and chewiness in beef frankfurters, due to the presence of pectin and fiber in the tomatoes; pectin could increase the shear value, while lignin and cellulose increase the hardness. However, the hardness of PBMAs decreased when the carrot powder was added. This may be due to the higher moisture content of carrot powder (18.87%), compared to caramel powder (2.8%) and tomato powder (5.79%). Wi et al. [55] reported that the higher water content in the meat analog decreases the hardness, cohesiveness, and chewiness of the product.

## 4. Conclusions

This study demonstrated that adding both carrot and tomato powders enhanced the quality characteristics of SPMPs compared to caramel powder. Notably, tomato powder showed superior efficacy in this regard. The incorporation of tomato and carrot powder increased the a* (redness), and b* (yellowness) values of the SPMPs, respectively, while the values decreased over storage, possibly due to the degradation of the pigments. The color stability of caramel SPMPs is the best during cooking; the ΔE of tomato SPMPs is lower than carrots, indicating that tomato SPMPs experienced less of a color change than carrots during cooking. Carrot SPMPs during refrigeration storage experienced less color stability and more color change. The addition of carrot and tomato powders resulted in a decrease in the pH level, which may enhance the storage stability of the SPMPs. Both carrot and tomato powders promoted the TPC and antioxidant stability of SPMPs and inhibited the lipid oxidation of SPMPs during cooking, while tomato powder exhibited superior efficacy in TPC and antioxidant activity compared to the others. The TPC of carrot and tomato SPMPs was relatively thermally stable, suggesting potentially more health benefits. Only the addition of tomato powder resulted in a decreasing trend of cooking loss. In the raw SPMPs, the addition of caramel and carrot pigments resulted in decreased hardness, but tomato powder increased the hardness. After cooking, the springiness of the SPMPs increased but cohesiveness and chewiness decreased with more tomato powder addition; the effect of the concentration of caramel and carrot powders on the texture characteristics was less pronounced in the cooked SPMPs. Sensory evaluation is suggested to further evaluate the impact of the three colorants on consumer acceptability of SPMPs.

## Figures and Tables

**Figure 1 foods-13-02224-f001:**
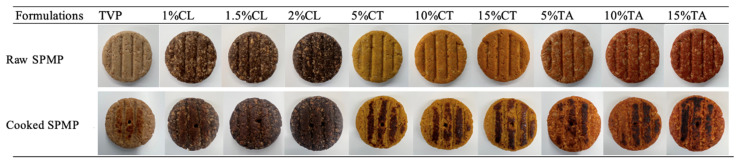
The appearance of raw and cooked SPMPs with different concentrations of various pigments. For sample codes, refer to Table 1.

**Table 1 foods-13-02224-t001:** The formulations of soy protein-based meat patties.

Ingredients	Formulations									
	TVP	1%CL	1.5%CL	2%CL	5%CT	10%CT	15%CT	5%TA	10%TA	15%TA
Basic ingredients (%)										
Textured vegetable protein	25.8	25.8	25.8	25.8	25.8	25.8	25.8	25.8	25.8	25.8
Boiled water	54.2	54.2	54.2	54.2	54.2	54.2	54.2	54.2	54.2	54.2
Wheat gluten flour	12	12	12	12	12	12	12	12	12	12
Coconut oil	6	6	6	6	6	6	6	6	6	6
Methylcellulose	1	1	1	1	1	1	1	1	1	1
Salt	1	1	1	1	1	1	1	1	1	1
Treatment (additional ingredient %)										
Caramel powder	-	1	1.5	2	-	-	-	-	-	-
Carrot powder	-	-	-	-	5	10	15	-	-	-
Tomato powder	-	-	-	-	-	-	-	5	10	15

**Table 2 foods-13-02224-t002:** The L* (lightness), a* (redness), and b* (yellowness) values of raw and cooked SPMPs (mean ± SD, *n* = 18).

Formulations	Raw SPMP			Cooked SPMP			
	L*	a*	b*	L*	a*	b*	ΔE_c_
TSP	64.09 ± 1.959 ^a,^*	4.94 ± 0.456 ^h,^*	20.68 ± 0.996 ^e,^*	55.00 ± 2.772 ^a,^*	5.95 ± 1.818 ^e,f,g,^*	27.59 ± 2.128 ^c,^*	11.75 ± 2.814 ^b,c^
1% CL	40.53 ± 1.427 ^f,^*	5.91 ± 0.357 ^g,^*	16.00 ± 0.804 ^f,^*	32.19 ± 1.733 ^e,f,^*	6.92 ± 0.959 ^d,e,^*	17.62 ± 1.540 ^d,^*	8.91 ± 2.076 ^a,b^
1.5% CL	34.64 ± 1.837 ^g,^*	5.98 ± 0.296 ^g,NS^	13.89 ± 0.771 ^g,^*	28.72 ± 1.825 ^f,g,^*	5.50 ± 0.972 ^e,f,g,NS^	14.83 ± 1.504 ^d,^*	6.46 ± 2.268 ^a^
2% CL	31.42 ± 1.580 ^h,^*	5.38 ± 0.394 ^g,h,NS^	11.83 ± 0.928 ^h,^*	26.15 ± 1.570 ^g,^*	4.99 ± 1.299 ^f,NS^	13.01 ± 1.676 ^d,^*	5.94 ± 1.158 ^a^
5% CT	58.66 ± 1.109 ^b,^*	8.05 ± 0.632 ^f,^*	40.28 ± 1.374 ^a,^*	39.28 ± 5.240 ^c,^*	10.43 ± 1.515 ^c,^*	35.59 ± 8.640 ^a,b,^*	21.11 ± 8.234 ^e^
10% CT	54.17 ± 1.295 ^c,^*	13.85 ± 0.576 ^e,^*	38.91 ± 1.099 ^b,NS^	45.14 ± 3.786 ^b,^*	8.48 ± 2.140 ^d,^*	37.92 ± 3.725 ^a,NS^	11.79 ± 2.812 ^b,c^
15% CT	53.20 ± 1.650 ^c,^*	16.42 ± 0.585 ^c,^*	40.12 ± 1.134 ^a,^*	38.51 ± 6.160 ^c,^*	10.55 ± 1.885 ^c,^*	33.07 ± 8.280 ^a,b,^*	18.80 ± 7.457 ^d,e^
5% TA	52.62 ± 1.670 ^c,^*	15.46 ± 1.079 ^d,NS^	29.02 ± 1.058 ^c,^*	39.99 ± 4.690 ^c,^*	15.74 ± 0.952 ^b,NS^	32.97 ± 4.061 ^b,^*	14.30 ± 4.101 ^c,d^
10% TA	48.18 ± 1.461 ^d,^*	17.49 ± 1.480 ^b,NS^	28.32 ± 0.773 ^c,d,^*	36.66 ± 3.214 ^c,d,^*	17.27 ± 1.085 ^a,b,NS^	31.43 ± 3.540 ^b,c,^*	12.83 ± 2.182 ^b,c^
15% TA	44.47 ± 1.518 ^e,^*	20.04 ± 0.963 ^a,^*	27.74 ± 1.005 ^d,NS^	33.51 ± 5.100 ^d,e,^*	17.37 ± 1.597 ^a,^*	27.74 ± 4.066 ^c,NS^	12.25 ± 5.102 ^b,c^

The different lowercase (a–h) superscripts in the same column denote a significant difference between formulations (*p* < 0.05). “*” indicates there is a significant difference between the raw and cooked SPMPs of the same formulation; “NS” indicates there is no significant difference between the raw and cooked SPMPs of the same formulation.

**Table 3 foods-13-02224-t003:** Changes in L* (lightness), a* (redness), and b* (yellowness) values of raw SPMP during 10 days of storage at 4 °C (mean ± SD, *n* = 18).

Parameter	Formulations	Day 0	Day 1	Day 4	Day 7	Day 10
**L***	TSP	63.08 ± 1.656 ^a,A^	64.03 ± 1.480 ^a,A^	63.52 ± 1.420 ^a,A^	63.06 ± 1.652 ^a,A^	63.78 ± 1.138 ^a,A^
	1%CL	40.48 ± 1.772 ^g,A^	40.98 ± 1.670 ^g,A^	41.19 ± 1.256 ^g,A^	40.67 ± 1.298 ^f,A^	41.46 ± 1.060 ^f,A^
	1.5%CL	36.13 ± 1.609 ^h,A^	35.94 ± 1.852 ^h,A^	35.13 ± 1.433 ^h,A^	35.96 ± 2.240 ^g,A^	36.23 ± 1.558 ^g,A^
	2%CL	32.43 ± 1.497 ^i,A^	32.25 ± 1.550 ^i,A^	31.66 ± 1.446 ^i,A^	32.57 ± 1.304 ^h,A^	32.47 ± 2.089 ^h,A^
	5%CT	58.06 ± 1.316 ^b,A^	58.18 ± 1.469 ^b,A^	57.71 ± 0.984 ^b,A^	58.32 ± 0.986 ^b,A^	58.24 ± 1.190 ^b,A^
	10%CT	54.55 ± 1.127 ^c,A^	53.90 ± 1.708 ^c,A,B,C^	52.45 ± 1.136 ^c,C^	53.07 ± 0.730 ^c,B,C^	54.28 ± 1.552 ^c,A,B^
	15%CT	52.42 ± 1.229 ^d,A^	52.81 ± 0.774 ^c,d,A^	52.69 ± 1.102 ^c,d,A^	53.13 ± 0.864 ^c,A^	53.06 ± 1.182 ^c,A^
	5%TA	50.90 ± 2.064 ^d,B^	51.37 ± 1.394 ^d,A,B^	50.41 ± 2.050 ^d,B^	51.79 ± 2.052 ^c,A,B^	53.00 ± 1.996 ^c,A^
	10%TA	46.69 ± 1.193 ^e,C^	47.18 ± 1.866 ^e,C,B^	47.34 ± 1.913 ^e,A,B,C^	48.66 ± 1.171 ^d,A^	48.30 ± 1.533 ^d,A,B^
	15%TA	43.33 ± 1.423 ^f,B^	43.25 ± 1.262 ^f,B^	44.13 ± 0.839 ^f,A,B^	43.88 ± 0.677 ^e,A,B^	44.69 ± 0.865 ^e,A^
**a***	TSP	4.92 ± 0.481 ^h,B^	5.25 ± 0.422 ^g,A,B^	5.39 ± 0.507 ^f,A,B^	5.56 ± 0.642 ^g,A^	5.53 ± 0.570 ^g,A^
	1%CL	5.86 ± 0.347 ^g,B^	6.62 ± 0.477 ^f,A^	6.69 ± 0.369 ^e,A^	6.64 ± 0.375 ^f,A^	6.88 ± 0.410 ^f,A^
	1.5%CL	6.01 ± 0.295 ^g,C^	6.31 ± 0.316 ^f,B,C^	6.38 ± 0.313 ^e,A,B^	6.49 ± 0.273 ^f,A,B^	6.63 ± 0.368 ^f,A^
	2%CL	5.34 ± 0.404 ^g,h,C^	6.13 ± 0.524 ^f,B^	6.22 ± 0.399 ^e,B^	6.30 ± 0.479 ^f,g,A,B^	6.63 ± 0.394 ^f,A^
	5%CT	8.05 ± 0.671 ^f,B^	7.48 ± 0.366 ^e,C^	8.54 ± 0.605 ^d,A,B^	8.17 ± 0.568 ^e,B^	8.93 ± 0.472 ^e,A^
	10%CT	13.88 ± 0.576 ^e,B,C^	12.97 ± 1.057 ^d,C^	14.21 ± 1.025 ^c,A,B^	15.32 ± 1.371 ^c,A^	14.20 ± 0.882 ^c,B^
	15%CT	16.54 ± 0.409 ^c,C^	15.47 ± 0.448 ^c,D^	17.25 ± 0.836 ^b,A,B^	16.53 ± 0.964 ^b,B,C^	17.36 ± 0.763 ^b,A^
	5%TA	15.59 ± 1.036 ^d,A^	14.94 ± 0.897 ^c,A^	14.68 ± 0.779 ^c,A,B^	13.84 ± 0.584 ^d,B,C^	13.40 ± 0.820 ^d,C^
	10%TA	17.57 ± 1.503 ^b,A^	18.10 ± 0.995 ^b,A^	17.87 ± 0.754 ^b,A^	17.30 ± 0.760 ^b,A^	17.19 ± 0.896 ^b,A^
	15%TA	20.18 ± 0.824 ^a,A,B^	20.72 ± 0.897 ^a,A^	19.65 ± 0.936 ^a,B^	19.86 ± 0.897 ^a,B^	19.51 ± 0.821 ^a,B^
**b***	TSP	20.65 ± 1.055 ^e,A^	20.03 ± 0.713 ^d,A^	20.10 ± 0.700 ^d,A^	20.01 ± 0.750 ^e,A^	20.33 ± 0.628 ^e,A^
	1%CL	15.98 ± 0.849 ^f,C^	16.59 ± 1.074 ^e,B,C^	16.75 ± 0.785 ^e,A,B,C^	17.19 ± 0.973 ^f,A,B^	17.45 ± 0.831 ^f,A^
	1.5%CL	13.89 ± 0.793 ^g,C^	14.42 ± 0.817 ^f,B,C^	14.30 ± 0.643 ^f,C^	15.09 ± 0.897 ^g,A,B^	15.49 ± 0.953 ^g,A^
	2%CL	11.75 ± 0.952 ^h,B^	12.84 ± 1.159 ^g,A^	13.08 ± 0.953 ^f,A^	13.18 ± 1.216 ^h,A^	13.86 ± 1.325 ^h,A^
	5%CT	40.29 ± 1.441 ^a,A^	38.75 ± 1.342 ^a,B^	35.67 ± 1.352 ^a,C^	34.81 ± 1.192 ^a,C,D^	34.11 ± 1.118 ^a,D^
	10%CT	38.98 ± 1.100 ^b,A^	37.40 ± 1.805 ^b,B^	33.04 ± 1.223 ^b,C,D^	33.61 ± 1.016 ^b,C^	32.41 ± 0.929 ^b,D^
	15%CT	40.31 ± 0.891 ^a,A^	37.23 ± 0.872 ^b,B^	33.19 ± 1.205 ^b,C^	34.05 ± 0.897 ^a,b,C^	33.06 ± 0.947 ^b,C^
	5%TA	29.15 ± 1.038 ^c,A^	27.62 ± 0.991 ^c,B^	27.11 ± 1.102 ^c,B^	27.17 ± 1.094 ^d,B^	27.40 ± 1.062 ^d,B^
	10%TA	28.30 ± 0.768 ^c,d,A,B^	27.63 ± 1.093 ^c,A,B^	26.58 ± 4.610 ^c,B^	28.79 ± 0.781 ^c,A^	29.25 ± 0.900 ^c,A^
	15%TA	27.67 ± 1.006 ^d,B,C^	26.73 ± 1.007 ^c,D^	26.83 ± 0.695 ^c,C,D^	28.26 ± 0.883 ^c,A,B^	28.98 ± 0.820 ^c,A^
**ΔE_s_**	TSP	-	2.19 ± 1.079 ^a^	2.46 ± 1.448 ^a^	2.67 ± 1.279 ^a^	2.54 ± 1.207 ^a^
	1%CL	-	2.69 ± 1.040 ^a^	2.54 ± 1.469 ^a^	2.48 ± 0.867 ^a^	2.86 ± 1.062 ^a^
	1.5%CL	-	2.37 ± 1.280 ^a^	2.59 ± 1.330 ^a^	2.94 ± 1.466 ^a^	2.54 ± 0.939 ^a^
	2%CL	-	2.49 ± 1.578 ^a^	3.00 ± 1.843 ^a^	2.35 ± 1.602 ^a^	3.28 ± 1.530 ^a^
	5%CT	-	2.81 ± 1.252 ^a^	5.09 ± 1.627 ^b,c^	5.85 ± 1.651 ^b^	6.55 ± 1.849 ^b^
	10%CT	-	3.26 ± 1.956 ^a^	6.53 ± 1.366 ^c,d^	6.11 ± 1.388 ^b^	6.97 ± 1.518 ^b^
	15%CT	-	3.57 ± 0.962 ^a^	7.41 ± 1.124 ^d^	6.47 ± 0.817 ^b^	7.50 ± 1.176 ^b^
	5%TA	-	2.77 ± 1.450 ^a^	3.74 ± 1.349 ^a,b^	3.51 ± 1.596 ^a^	3.98 ± 1.958 ^a^
	10%TA	-	2.86 ± 1.661 ^a^	3.46 ± 4.326 ^a,b^	2.75 ± 1.694 ^a^	2.89 ± 0.876 ^a^
	15%TA	-	2.84 ± 1.018 ^a^	2.33 ± 0.994 ^a^	2.15 ± 0.800 ^a^	2.94 ± 0.883 ^a^

The different lowercase (a–i) superscripts in the same column denote a significant difference between formulations (*p* < 0.05). The different uppercase (A–D) superscripts in the same row denote a significant difference between storage days (*p* < 0.05).

**Table 4 foods-13-02224-t004:** Changes in pH values of raw SPMP during 10 days of storage at 4 °C (mean ± SD, *n* = 3).

Formulations	pH				
	Day 0	Day 1	Day 4	Day 7	Day 10
TSP	6.72 ± 0.034 ^A,a^	6.43 ± 0.010 ^B,a^	6.30 ± 0.053 ^C,a^	6.25 ± 0.071 ^C,a^	6.21 ± 0.010 ^C,b^
1% CL	6.51 ± 0.004 ^A,b^	6.37 ± 0.004 ^B,b^	6.30 ± 0.010 ^C,a^	6.29 ± 0.013 ^C,D,a^	6.28 ± 0.006 ^D,a^
1.5% CL	6.51 ± 0.001 ^A,b^	6.35 ± 0.010 ^B,c^	6.24 ± 0.010 ^D,a^	6.28 ± 0.003 ^C,a^	6.26 ± 0.006 ^C,a^
2% CL	6.47 ± 0.002 ^A,b^	6.34 ± 0.010 ^B,c^	6.27 ± 0.006 ^C,a^	6.26 ± 0.006 ^D,a^	6.21 ± 0.006 ^E,b^
5% CT	6.38 ± 0.022 ^A,c^	6.26 ± 0.009 ^B,d^	6.30 ± 0.010 ^B,a^	6.14 ± 0.023 ^C,b^	6.17 ± 0.015 ^C,c^
10% CT	6.35 ± 0.006 ^A,c^	6.25 ± 0.003 ^C,d^	6.27 ± 0.000 ^B,a^	6.16 ± 0.007 ^D,b^	6.14 ± 0.000 ^E,c^
15% CT	6.27 ± 0.010 ^A,d^	6.20 ± 0.006 ^B,e^	6.18 ± 0.032 ^B,C,b^	6.14 ± 0.007 ^C,D,b^	6.13 ± 0.006 ^D,c^
5% TA	5.89 ± 0.009 ^A,e^	5.82 ± 0.013 ^B,f^	5.82 ± 0.012 ^B,c^	5.79 ± 0.008 ^B,C,c^	5.76 ± 0.021 ^C,d^
10% TA	5.46 ± 0.006 ^A,f^	5.39 ± 0.002 ^B,g^	5.29 ± 0.012 ^D,d^	5.34 ± 0.010 ^C,d^	5.23 ± 0.017 ^E,e^
15% TA	5.18 ± 0.001 ^A,g^	5.11 ± 0.003 ^B,h^	4.95 ± 0.006 ^E,e^	5.07 ± 0.007 ^C,e^	5.03 ± 0.021 ^D,f^

The different lowercase (a–h) superscripts in the same column denote a significant difference between formulations (*p* < 0.05). The different uppercase (A–E) superscripts in the same row denote a significant difference between storage days (*p* < 0.05).

**Table 5 foods-13-02224-t005:** The total phenolic content (mg gallic acid/100 g sample), DPPH radical scavenging activity (RSA%), and TBARS values (mg MDA/kg sample) of raw and cooked SPMPs (mean ± SD, *n* = 3).

	TPC (mg GAE/100 g)		RSA (%)		TBARS (mg MDA/kg Sample)	
Formulations	Raw SPMP	Cooked SPMP	Raw SPMP	Cooked SPMP	Raw SPMP	Cooked SPMP
TSP	58.74 ± 0.526 ^e,f,NS^	58.34± 2.080 ^e,f,NS^	22.46 ± 7.770 ^d,f,NS^	34.66 ± 1.696 ^e,NS^	0.67 ± 0.035 ^g,^*	2.78 ± 0.111 ^c,^*
1% CL	54.21 ± 1.199 ^g,NS^	56.85 ± 1.551 ^e,f,NS^	21.16 ± 7.120 ^d,f,^*	36.79 ± 5.120 ^e,^*	0.69 ± 0.024 ^g,^*	2.78 ± 0.123 ^c,^*
1.5% CL	55.48 ± 2.610 ^e,f,g,NS^	57.44 ± 2.140 ^e,f,NS^	22.29 ± 6.590 ^d,f,^*	36.37 ± 4.710 ^e,^*	0.68 ± 0.055 ^g,^*	2.82 ± 0.084 ^c,^*
2% CL	55.25± 1.237 ^f,g,NS^	55.68 ± 0.398 ^f,NS^	23.81 ± 6.520 ^d,e,f,^*	35.03 ± 5.020 ^e,^*	0.68 ± 0.028 ^g,^*	2.80 ± 0.037 ^c,^*
5% CT	59.24± 0.360 ^e,NS^	60.32 ± 0.831 ^e,NS^	39.42 ± 4.480 ^c,e,^*	43.42 ± 4.120 ^d,e,^*	1.33 ± 0.016 ^f,^*	2.76 ± 0.122 ^c,^*
10% CT	63.94± 0.642 ^d,NS^	64.52 ± 0.969 ^d,NS^	47.30 ± 1.294 ^c,NS^	53.09 ± 4.430 ^c,d,NS^	2.03 ± 0.014 ^d,^*	3.45 ±0.090 ^b,^*
15% CT	69.91± 1.690 ^c,NS^	68.55 ± 0.969 ^c,NS^	52.69 ± 3.990 ^c,^*	59.89 ± 3.550 ^b,c,^*	2.32 ± 0.032 ^c,^*	3.87 ± 0.162 ^a,^*
5% TA	68.49 ± 1.564 ^c,NS^	69.71 ± 1.037 ^c,NS^	53.44 ± 4.180 ^b,c,NS^	55.51 ± 2.920 ^b,c,NS^	1.69 ± 0.032 ^e,^*	2.28 ± 0.035 ^d,^*
10% TA	74.41± 0.509 ^b,NS^	74.27 ± 0.437 ^b,NS^	67.50 ± 0.555 ^b,NS^	65.55 ± 0.304 ^b,NS^	2.52 ± 0.017 ^b,^*	3.28 ± 0.050 ^b,^*
15% TA	85.86 ± 1.044 ^a,NS^	85.63 ±0.419 ^a,NS^	82.27 ± 1.950 ^a,^*	84.16 ± 1.890 ^a,^*	3.18 ± 0.031 ^a,^*	4.16 ± 0.226 ^a,^*

The different lowercase (a–g) superscripts in the same column denote a significant difference between formulations (*p* < 0.05). “*” indicates there is a significant difference between the raw and cooked SPMPs of the same formulation; “NS” indicates there is no significant difference between the raw and cooked SPMPs of the same formulation.

**Table 6 foods-13-02224-t006:** The cooking loss (%) and texture profiles of raw SPMPs (mean ± SD, *n* = 9).

Formulations	Cooking Loss (%)	Hardness (N)		Cohesiveness		Springiness (mm)		Chewiness (N)	
		Raw	Cooked	Raw	Cooked	Raw	Cooked	Raw	Cooked
TSP	8.24 ± 0.174 ^a,b^	9.52 ± 0.91 ^b,^*	16.87 ± 0.82 ^c,^*	0.29 ± 0.03 ^a,^*	0.34 ± 0.03 ^a,^*	3.72 ± 0.36 ^a,^*	2.17 ± 0.16 ^a,^*	1.50 ± 0.41 ^a,^*	4.12 ± 0.57 ^a,b,^*
1% CL	9.65 ± 0.305 ^a,b^	8.36 ± 0.40 ^c,^*	17.76 ± 1.75 ^b,c,^*	0.22 ± 0.011 ^c,d,e,^*	0.33 ± 0.02 ^a,^*	4.35 ± 0.59 ^a,^*	2.18 ± 0.33 ^a,^*	0.78 ± 0.10 ^d,e,^*	4.11 ± 0.79 ^a,b,^*
1.5% CL	8.60 ± 2.080 ^a,b^	7.63 ± 0.67 ^c,d,^*	16.71 ± 2.51 ^c,d,^*	0.21 ± 0.03 ^d,e,^*	0.31 ± 0.03 ^a,b,^*	4.14 ± 1.17 ^a,^*	2.37 ± 0.57 ^a,^*	0.73 ± 0.28 ^d,e,^*	3.61 ± 0.98 ^b,c,^*
2% CL	7.36 ± 0.864 ^b^	7.42 ± 0.34 ^d,e,^*	14.10 ± 1.66 ^d,e,^*	0.20 ± 0.01 ^e,^*	0.28 ± 0.02 ^b,c,^*	4.11 ± 0.97 ^a,^*	2.65 ± 0.83 ^a,b,^*	0.63 ± 0.08 ^e,^*	2.68 ± 0.64 ^c,d,^*
5% CT	9.11 ± 1.101 ^a,b^	7.73 ± 0.40 ^c,d,^*	16.61 ± 1.22 ^c,d,^*	0.23 ± 0.01 ^c,d,^*	0.31 ± 0.02 ^a,b,^*	4.40 ± 0.19 ^a,^*	2.53 ± 0.28 ^a,b,^*	0.76± 0.10 ^d,e,^*	3.42 ± 0.54 ^b,c,^*
10% CT	10.74 ± 0.648 ^a^	7.45 ± 0.42 ^d,e,^*	17.22 ± 1.60 ^c,^*	0.25 ± 0.02 ^b,c,^*	0.31 ± 0.02 ^a,b,^*	4.12 ± 0.16 ^a,^*	2.56 ± 0.32 ^a,b,^*	0.79 ± 0.06 ^d,e,^*	3.43 ± 0.65 ^b,c,^*
15% CT	8.19 ± 0.587 ^a,b^	6.68 ± 0.36 ^e,^*	12.10 ± 0.81 ^e,^*	0.24 ± 0.01 ^b,c,^*	0.26 ± 0.01 ^c,d,^*	4.22 ± 0.25 ^a,^*	3.43 ± 0.32 ^c,^*	0.65 ± 0.07 ^e,^*	1.66 ± 0.25 ^e,^*
5% TA	10.58 ± 0.956 ^a^	11.19 ± 0.54 ^a,^*	20.92 ± 2.83 ^a,^*	0.26 ± 0.01 ^b,^*	0.31 ± 0.02 ^a,b,^*	3.93 ± 0.20 ^a,^*	2.12 ± 0.22 ^a,^*	1.38 ± 0.14 ^a,b,^*	4.56 ± 0.73 ^a,^*
10% TA	7.81 ± 0.305 ^b^	11.52 ± 0.36 ^a,^*	19.95 ± 1.57 ^a,b,^*	0.22 ± 0.01 ^c,d,e,^*	0.26 ± 0.02 ^c,d,^*	4.23 ± 0.16 ^a,^*	2.99 ± 0.16 ^b,c,^*	1.12 ± 0.08 ^b,c,^*	3.07 ± 0.20 ^c,d,^*
15% TA	7.19 ± 0.770 ^b^	11.30 ± 0.56 ^a,^*	19.13 ± 1.12 ^a,b,c,^*	0.21 ± 0.02 ^d,e,^*	0.23 ± 0.01 ^d,^*	4.38 ± 0.28 ^a,^*	3.43 ± 0.16 ^c,^*	0.98 ± 0.16 ^c,d,^*	2.30 ± 0.22 ^d,e,^*

The different lowercase (a–e) superscripts in the same column denote a significant difference between formulations (*p* < 0.05). “*” indicates there is a significant difference between the raw and cooked SPMPs of the same formulation.

## Data Availability

The original contributions presented in the study are included in the article, further inquiries can be directed to the corresponding author.

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
