# Peer review of "Effects of Incorporating Caramel, Carrot, and Tomato Powder on the Quality Characteristics of Soy Protein-Based Meat Patties"

_foods, 2024, doi:10.3390/foods13142224_

Round 1
Reviewer 1 Report
Comments and Suggestions for Authors
This is not a paper on complete meat substitutes, but a study on the differences in the basic ingredients used to make patties. The analysis is also lacking. The paper is also not creative at all.
This is a study on the quality effects of adding caramel, carrot, and tomato powder to soy protein patties, respectively. It is judged that the paper lacks creativity.
1. The analysis method includes weight, height, general component analysis (moisture, crude protein, ash, and fat), total flavonoids, ABTS radical scavenging activity, and preference test when caramel, carrot, and tomato powder are added to each meat patty. I think it should be done. In addition, essential and non-essential amino acids, fatty acids, general bacteria, and E. coli should also be analyzed to improve the quality of the paper.
2. The analysis method currently presented in this paper is considered to be insufficient in that it is a very basic analysis method and is submitted to an international journal. In other words, it is not enough to see the effect of adding caramel, carrot, and tomato powder to the meat patty.
3. In the discussion section, specific figures must be presented and compared with references.
4. In Table 1, you must provide the mixing ratio of the ingredients in detail.
5. Fig. 1. Do not present only the outer surface; photos of the inside of the patty must also be presented for a clear comparison.
6. The conclusion part of the paper is not clear.
Overall, the expression of validity of the paper is insufficient.
Comments on the Quality of English LanguageEnglish is generally understandable.
Reviewer 2 Report
Comments and Suggestions for Authors
the lack of line numbers significantly complicates the reviewing process!!!
The subject matter of the work is very interesting and reflects both cultural, economic and ecological trends.
thesis topic, I suggest "soy protein-based meat" over "soy meat analogues"
keywords add "meat analogues"
"2.2.1. Preparation of soy protein meat patties (SPMPs)
The uncolored meat patties" - the word meat is misleading, it suggests changing it to either patties or meat analogues
in table 1, to make it more readable, I suggest replacing "0" with "-"
2.3. Color measurement - why was calibration:l" not 100? why is delta E not determined when measuring color - as an indicator differentiating control samples from samples with added caramel, carrot or tomato?
I think that an important parameter that should be discussed in the work is the change in increase/decrease in nutritional value, but I assume that the authors, as they indicated in the conclusions, will perform this research alongside the analysis of consumer acceptance
Reviewer 3 Report
Comments and Suggestions for Authors
I think that the article entitled “Effects of incorporating caramel, carrot, and tomato powder on the quality characteristics of soy protein-based meat patties” presents interesting results related to the use or benefit of alternative proteins. However, some adjustments are necessary in the manuscript.
The abstract must be rewritten, I think that in this section the most significant results of the work should be highlighted. In the abstract of this work a lot of general information is observed.
I think the key words are appropriate.
Introduction
I think that the introduction should be supplemented with key information about the job, for example.
Quantity of water used in the production of meat of animal origin (bovine, pork and pig), compared with soy protein.
Quantity of greenhouse gases emitted in meat production.
What types of cancer are there? “Some forms of cancers?”
Market expansion for alternative protein sources
I think that the materials and methods are adequate
Results and discussion
Color measurement
I think that this section should be enriched with more information, it is necessary to compare the results with other published works, and to explain the possible phenomena. In dry pH measurement, Total phenolic content (TPC), DPPH radical scavenging activity, Lipid oxidation (TBARS assay), Cooking loss, and Texture profile analysis also observed - discussions of the results are fine, please improve.
Finally adjust the conclusion of the manuscript.
The references are adequate
Reviewer 4 Report
Comments and Suggestions for Authors
Title : Effects of incorporating caramel, carrot, and tomato powder on the quality characteristics of soy protein-based meat patties
The aim of this study is to evaluate the effect of incorporating caramel, carrot, and tomato powder on the quality characteristics of soy protein-based meat patties. This topic is particularly relevant in the context of the growing importance of plant proteins in the human diet. The measured indices include color, rheological properties (such as cooking yield, hardness, cohesiveness, springiness, and chewiness), and nutritional quality indices (such as total polyphenol content and antioxidant activity). While this study is interesting and valuable,
1. I believe it would benefit from also evaluating the sensory quality of the patties, which is crucial for assessing the acceptability of this food product.
2. The study's abstract needs significant improvement, including a comparative analysis of the effects of caramel, carrot, and tomato powder on the quality characteristics of soy protein-based meat patties. The revised abstract should describe the impact of these three pigments on the patties' rheological properties, color, and nutritional quality.
3. Why was caramel chosen, given that it is known to be toxic and lacks beneficial health effects?
4. it is insufficient to measure only total polyphenols to assess nutritional quality.
5. The introduction is poorly written and poorly organized. It is necessary to start by presenting the importance of plant proteins in the human diet. Then, discuss the significance of the pigments used in this study, highlighting their nutritional and industrial importance.
6. Section 2.2.1, the justification for choosing these doses needs to be explained.
Round 2
Reviewer 1 Report
Comments and Suggestions for Authors
I have checked the contents carefully.
The content has been well edited, including color differences.
thank you